# Prevalence of gastroesophageal reflux disease and its associated risk factors among the adult population of Mthatha, Eastern Cape, South Africa

Nomonde Ndyalvan, Mirabel K. Nanjoh◉*, Monwabisi Faleni, Wezile W. Chitha, Sibusiso C. Nomatshila

Department of Public Health, Faculty of Medicine and Health Sciences, Walter Sisulu University, Mthatha, Eastern Cape, South Africa

* mnanjoh@wsu.ac.za

## Abstract

Epidemiological data on gastroesophageal reflux disease (GERD) reveal variations in prevalence rates and risk factors, which are still to be elucidated in the South African setting. Population-specific data are needed to contribute to the understanding of the pathophysiology of GERD. This study aims to determine GERD's prevalence and associated risk factors among Mthatha residents. A descriptive cross-sectional study was carried out with a systematically sampled 353 adults attending the outpatient department of a secondary health care facility using the standardized GERD questionnaire (GerdQ) for screening and diagnosis. Descriptive and inferential statistics were performed with the aid of SPSS version 26. The prevalence of GERD was 32.0 percent (CI: 27.3% to 37.0%). Only 6.5 percent had an 89% likelihood of having GERD. Over half (53.1%) had GERD with little effect on day-to-day activities. The most common symptoms were heartburn at 49.6 percent, sleep disturbance at 34.8 percent, chest pain at 32.3 percent, and regurgitation at 32.3 percent, and were mild in most of them. Hypertension (OR = 3.0, CI: 1.5–6.4, p = 0.003) and eating more than three meals a day (OR = 4.2, CI: 1.9–9.2, p < 0.0001) were the major risk factors for GERD. The GERD rate of one in every three people is higher than the global rate of one in every six. The association of GERD with hypertension requires consideration of screening for GERD in patients with hypertension. Community education on proper dietary habits and the consequences of excess daily meals is needed to interrupt the development of GERD in the current population.

## Introduction

Gastroesophageal reflux disease (GERD) is a chronic disorder of the upper gastro-intestinal tract characterized by disturbing symptoms that develop from the backflow

**Data availability statement:** The datasets generated and analyzed in this study are within the Supporting information file.

**Funding:** The author(s) received no specific funding for this work.

**Competing interests:** The authors have declared that no competing interests exist.

of stomach contents into the esophagus [1–3]. GERD ranks as the most prevalent gastrointestinal-related diagnosis in medical settings [1–5] and is associated with reduced health-related quality of life [6,7] and disability [8,9]. The disorder raises the chance of Barrett's esophagus, erosive esophagitis, and esophageal cancer if treatment is not initiated timeously [10–12].

According to available pathophysiology literature, the pathogenesis of GERD is not fully understood. Undoubtedly, intrinsic, structural, or both mechanisms contribute to the onset of this disorder [13]. GERD is believed to be caused by a defect at the gastroesophageal junction, defective esophageal peristalsis, transient relaxations of the lower esophageal sphincter, abnormalities in lower esophageal sphincter pressure, or impaired esophageal bolus transit [11,14]. GERD is also caused by a hiatal hernia, which also lowers lower sphincter pressure and delays stomach emptying [11,15]. The development of GERD is thought to be influenced by esophageal visceral hypersensitivity as well [11,15].

Whatever physiological changes, it allows for regularly evacuating stomach contents such as gastric acid, pepsin, and bile into the esophagus [16]. GERD symptoms include heartburn, regurgitation, and epigastric pain [3,17]. The symptoms can manifest outside the esophagus as dysphagia, odynophagia, belching, nausea, chest pain, asthma, persistent cough, laryngitis, tooth erosions, and occasionally tissue lesions [18–21].

In 2019, there were 783.95 million cases of GERD worldwide [8]. The global pooled prevalence was 13.98% (95% CI: 12.47% to 15.56%) in 2017 [22]. North America had the highest regional pooled prevalence of 19.55 percent, while Latin America and the Caribbean had the lowest of 12.88 percent [22]. The prevalence varies by country, ranging from 4.16 percent in China, 22.4 percent in Turkey [22] to 20.6 to 67.8 percent in Saudi Arabia [2,4]. The prevalence rate in African settings ranges from 32.1 percent in Ethiopia [23], 7.6 percent in Nigeria [24] to a pooled prevalence rate of 15 to 19.9 percent in West African countries [22]. According to two studies, the prevalence of GERD in the South African population ranges from 27 to 40 percent [25,26].

Demographic and lifestyle factors, augmented in some cases by familial predisposition, are associated with a higher GERD risk. Findings from two systematic reviews document increasing age and obesity as risk factors associated with GERD [22,27]. Other demographic risk factors include female gender, low educational level, urban residence, low income, and divorced/separated/widowed [22]. Examples of lifestyle factors are alcohol consumption, smoking, a lack of physical activity, and poor nutrition [28–30]. There is also a 31 percent chance of familial heritability [31]. Family history significantly predicted GERD in Saudi Arabia [2,3]. Genetic factors are also linked to a higher risk of GERD, where reflux symptoms are present. Single-nucleotide polymorphisms at the C allele in FOXF1 rs9936833 (FOXF1), the A allele in MHC rs9257809 (MHC), and Cyclin D1 (CCND1) are examples [31]. DNA repair and anti-inflammatory cytokine genes are also strongly linked to a higher risk of GERD [31].

There has never been research on the prevalence of GERD in the Eastern Cape's rural communities. Literature on associated factors in the rural Eastern Cape is

scarce. Similar research on this topic was conducted among rural residents in the Eastern Cape province of South Africa [32]. However, the study that used 24-hour ambulatory esophageal impedance-pH monitoring only reported the type and extent of reflux in a rural African community [32]. Estimates on reflux prevalence were not reported, but the authors highlighted that mild acid reflux was common in the community [32].

Epidemiological studies are needed to discover etiologies and risk factors in different populations to understand this disease better [33]. Therefore, the present study aims to determine GERD's prevalence and associated risk factors among residents in Mthatha, a secondary city with predominantly rural communities in the Eastern Cape province of South Africa. Awareness of GERD's prevalence would help educate people about lifestyle choices that can help them manage GERD symptoms, enhance their quality of life, and avoid GERD complications. Identifying risk factors could help develop public health interventions for populations at high risk of GERD.

## Materials and methods

### Study design and period

A descriptive cross-sectional study design was utilized to estimate GERD's prevalence and identify significant risk factors among the adult population of Mthatha. The study was carried out over one month, from 8th October 2024–11th November 2024.

### Study setting

The study was conducted at the Mthatha Regional Hospital, a secondary care district hospital. The regional hospital is located in Mthatha, a secondary city in the King Sabata Dalindyebo sub-district municipality, which forms part of the five sub-districts of the OR Tambo District municipality. Mthatha Regional Hospital is a technologically advanced and well-equipped facility providing specialized healthcare services to patients from 20 healthcare facilities (11 community healthcare centres and nine district hospitals) under the auspices of the Eastern Cape Department of Health. Although it serves as a referral hospital for patients from other regions in the district municipality, most patients attended to are from over 20 neighborhoods within the Mthatha locality. The neighborhoods are a mix of urban (Fort Gale and Southernwood), peri-urban (Mandela Park, Sibangweni, and Payne), and rural (Ncise, Decoligny Mission, Tabase, and Corana). Therefore, the study setting provides healthcare services to a mix of patients from different communities around Mthatha, supporting the implementation of the choice design within the facility.

### Study population

The target population was all patients seeking medical attention at the Mthatha Regional Hospital's outpatient department (OPD) who are resident in Mthatha, an area with a population size of more than 220,000 people [34], Inclusion (both gender and adult 18 years above) and exclusion (pregnant women and selected comorbidity including a diagnosis or history of achalasia, esophagitis, bronchitis, peptic ulcers, a history of upper gastrointestinal tract surgery, scleroderma, or caustic injury) were taken into consideration.

### Sampling

The participants were chosen using a systematic random sampling technique. The sample frame comprised all 2500 outpatients (hospital database) booked for a hospital visit during the one-month study period. The first participants for each data collection day were recruited randomly, while the others were selected using a sampling constant of six, obtained by dividing the total number of units in the sample frame (N = 2500) by the expected maximum sample size (n = 406). This sampling method was more straightforward, considering the number of people sitting simultaneously at the OPD department. There is no sitting pattern at the OPD other than "first come, first sit," so this corrects any skewness in data that could arise if there were no such pattern, and systematic sampling applied.

## Sample size

Gastroesophageal reflux disease prevalence is expected to be 40 percent, as in the Western Cape [26], with an absolute precision of 5% and a 95% confidence interval. The sample size of 369 was obtained using the following formula $n = Z^2 * p * (1 - p)/e^2$ [35]. A 10 percent (n = 37) adjustment was made to account for missing data due to no response. As a result, the final expected maximum sample size is 406, and the minimum is 332. The sample size 353 reached during the data collection period is within the expected maximum and minimum sample size.

## Data collection tool and technique

Data was collected using a questionnaire, which included validated questionnaires (GerdQ) and variables from previous studies that could answer the study objectives. Reliability testing produces a Cronbach alpha coefficient of 77.6 percent, indicating that the items in the research tool measured the same underlying concept: GERD prevalence and associated risk factors. Three research assistants and the primary researcher completed each questionnaire in 30–45 minutes. The following demographic data were collected: age, gender, level of education, civil status, work status, average monthly income, and source of income. Clinical data includes weight and height, which were used to calculate the body mass index (BMI). Medical history, including comorbidities (hypertension, diabetes mellitus, hyperlipidemia, ischemic heart disease), were also collected. The gastroesophageal reflux questionnaire (GerdQ) developed by Jones et al. was used as the diagnostic tool for GERD in the study population [36]. At a cut-off GerdQ score of more than 8, the GerdQ has a sensitivity of 65% and a specificity of 71% for the diagnosis of GERD, comparable to that achieved by gastroenterologists [36]. Six items comprise this questionnaire, four of which are regarded as positive predictors: regurgitation, heartburn, sleep disturbance brought on by regurgitation or heartburn, and over-the-counter medications other than those prescribed for GERD. The other two items—nausea and epigastric pain—are negative predictors. The frequency of GERD symptoms throughout the preceding seven days determines the GerdQ score. For negative predictors, the scoring rate is flipped (3 being none) and is based on a Likert scale, where 0 means none, 1 means one day, 2 means two to three days, and three means four to seven days. After summarization, a GerdQ score of eight (8) or higher indicates that the participant has GERD. A score of less than eight (8) means a low probability of GERD. A modest effect on day-to-day living is indicated by a GerdQ score of ≥8, and a score of <3 for Questions 5 and 6. On Questions 5 and 6, a score of ≥3 and a GerdQ score of ≥8 indicate that GERD has a major effect on day-to-day living.

The questionnaire also includes lifestyle factors such as smoking, physical activities, exercise, and nutritional practices based on variables available in literature [2,4] to see if they are linked to GERD in this rural population. Dietary behaviors linked to GERD were identified using a 21-food item questionnaire. The questionnaire was pretested in a pilot study comprising 20 participants (5% of the sample size) to assess the comprehensiveness of the representation of required variables, relevance, uniformity, and precision of the questionnaire, before the actual study initiation.

## Data analysis

The outcome variable was the presence or absence of GERD, which was reported as counts and percentages. Continuous independent variables, including age and BMI, are reported as median and interquartile range, while categorical independent variables, including age groups, nutritional status, sex, residence, and others, are reported as counts and percentages. If assumptions were met, the association to the outcome variable was based on Chi-square or Fisher's exact p-value, and the strength of the association was based on the Cramer's V values. Binary logistic regression's 95% confidence intervals and adjusted odds ratios were presented. The Hosmer-Lemeshow test assessed the logistic regression model's fitness; a significant value of 0.266 was obtained, confirming the model's goodness. The Firth logistic regression was also computed to reduce bias in estimated odds ratios where small subgroup sizes exist. A p-value of 0.05 or

less defines the significance level of observed odds ratios. All statistical tests were executed using the statistical program IBM SPSS version 26 (SPSS, Inc., Chicago, IL).

### Ethical considerations

This research was approved and ethical clearance issued (Ref no: 159/2024) by the Faculty of Medicine and Health Sciences Research and Bioethics Committee. Approval to conduct the study was obtained from the Eastern Cape Department of Health and the study site management. The research was voluntary, with no participants being coerced and supported with a signed written informed consent form. No names or personal identifiers, such as a home address, ID number, contact information, or workplace address, appeared on the data collection sheet. Access to complete questionnaires and consent forms is restricted to only those involved in this investigation. No harm was intended, and none was reported by anyone participating in the study.

### Results

Out of the 353 participants, 113 (32.0% CI: 27.3% to 37.0%) were found to have GERD, while 240 (68.0% CI: 63.0% to 72.7%) had no GERD; that is, one out of every three participants was suffering from GERD. According to the likelihood of having GERD, 68.0 percent (CI: 63.0% to 72.7% n = 240) of the participants had a 50% likelihood, 25.5 percent (CI: 21.2% to 30.2% n = 90) of the participants had a 79% likelihood, and 6.5 percent (CI: 4.3% to 9.5% n = 23) of the participants had 89% likelihood. There was a 46.9 percent prevalence of GERD with a substantial impact on daily life (CI: 37.9% to 56.1% n = 53) and a 53.1 percent prevalence of GERD with a modest effect on daily life (CI: 43.9% to 62.1% n = 60). The most common symptom was heartburn (49.6%; 95% CI: 44.4% − 54.8% n = 175), followed by sleep disturbance (34.8%; 95% CI: 30.0% − 39.9% n = 123), chest pain (32.3%; 95% CI: 27.6% − 37.3% n = 114), and regurgitation (32.0%; 95% CI: 27.3% − 37.0% n = 113). GERD symptoms were mild in most participants.

Females were at a greater risk of having GERD compared to males (crude odds ratio = 1.8, 95% CI: 1.0–3.0; p 0.033). None of the other demographic factors was associated with GERD. However, the probability of having GERD was higher among participants aged 40–59 years, single, still to complete Grade 12, unemployed, and had an average monthly income of R 500.00 to R 1000.00 (Table 1).

Clinical factors associated with GERD were hypertension and a family history of GERD. A significantly higher proportion (43.9%, p = 0.031) of hypertensive participants had GERD compared to their non-hypertensive counterparts (29.3%). A significantly high proportion (43.7%, p = 0.027) of participants with a family history of GERD had GERD compared to those without familial predisposition (Table 2).

Lifestyle factors associated with GERD were alcohol consumption, walking less than 30 minutes per week, walking 30 minutes or more per week, swimming less than 30 minutes per week, exercising less than 30 minutes per week, frequency of alcohol consumption, and number of meals a day. A significantly high proportion of participants with GERD consumed alcohol (43.5% p = 0.003), drank alcohol more than three times a week (100.0% p < 0.0001), never walked for less than 30 minutes per week (46.7% p = 0.008), walked only once for 30 minutes or more per week (65.0% p = 0.004), swam only once for less than 30 minutes per week (100.0% p = 0.003), exercise once for less than 30 minutes per week (91.7% p < 0.0001), and consumed more than three meals a day (64.3% p < 0.0001) (Table 3).

GERD was prevalent among non-drinkers of citrus drinks (38.5% p = 0.030), non-users of milk (59.6% p < 0.0001), non-eaters of bread (52.9% p = 0.001), non-consumers of fruits (58.8% p = 0.001), and non-consumers of sugar (41.3% p = 0.0038) (Table 4).

A binary logistic regression model shows that GERD could be caused independently by hypertension and eating more than three meals a day. It is 3.0 times more likely to develop GERD if hypertensive Firth adjusted odds ratio (FaOR) = 3.0, 95% CI: 1.5–6.4, p = 0.003) and 4.2 times more likely if food consumption exceeds three meals per day (FaOR = 4.2, 95% CI: 1.9–9.2, p = 0 < 0001). Swimming less than 30 minutes per week (FaOR = 3.3, 95% CI: 0.9 to 13.4 p = 0.075), not consuming fruits (FaOR = 2.6, 95% CI: 0.9 to 7.8 p = 0.077), drinking alcohol (FaOR = 1.6, 95% CI: 0.9 to 2.9 p = 0.082), and

**Table 1. Demographic factors associated with GERD among the adult population of Mthatha, South Africa.**

| Variables of interest | Categories | GERD, n (%) N = 113 | No GERD, n (%) N = 240 | Cramer's V | p-value |
|---|---|---|---|---|---|
| Age Groups | <40 years | 58 (30.7) | 131 (69.3) | 0.112 | 0.110 |
| | 40-59 years | 45 (38.1) | 73 (61.9) | | |
| | ≥60 years | 10 (21.7) | 36 (78.3) | | |
| Sex | Male | 23 (23.5) | 75 (76.5) | 0.114 | 0.033 |
| | Female | 90 (35.3) | 165 (64.7) | | |
| Marital status | Single | 85 (36.8) | 146 (63.2) | 0.148 | 0.150* |
| | Cohabiting | 2 (25.0) | 6 (75.0) | | |
| | Married | 19 (21.3) | 70 (78.7) | | |
| | Divorced | 2 (25.0) | 6 (75.0) | | |
| | Widowed | 4 (33.3) | 8 (66.7) | | |
| | Separated | 1 (20.0) | 4 (80.0) | | |
| Level of education | Never attended school | 10 (27.0) | 27 (73.0) | 0.152 | 0.188* |
| | Dropped out of school | 37 (26.4) | 103 (73.6) | | |
| | Still to complete Grade 12 | 4 (66.7) | 2 (33.3) | | |
| | Completed Grade 12 | 33 (37.1) | 56 (62.9) | | |
| | Student at a tertiary institution | 18 (39.1) | 28 (60.9) | | |
| | Completed tertiary education | 10 (30.3) | 23 (69.7) | | |
| | Vocational training | 1 (50.0) | 1 (50.0) | | |
| Employment status | Unemployed | 92 (33.9) | 179 (66.1) | 0.076 | 0.363 |
| | Self-employed | 10 (25.0) | 30 (75.0) | | |
| | Employed | 11 (26.2) | 31 (73.8) | | |
| Income per month | < R500 | 47 (30.3) | 108 (69.7) | 0.077 | 0.551 |
| | R500-R1000 | 29 (39.2) | 45 (60.8) | | |
| | R1000-R2000 | 14 (30.4) | 32 (69.6) | | |
| | >R2000 | 23 (30.3) | 53 (69.7) | | |

*Fisher's Exact.

not eating bread (FaOR = 2.1, 95% CI: 0.9 to 4.8 p = 0.095) was associated with an increased likelihood of having GERD; however, the significance levels were above the threshold (p > 0.05) (Table 5).

## Discussion

In the present study, among the 353 sampled participants, 113 were found to have GERD, accounting for 32.0 percent of the population. The rate is lower than 40% in the Western Cape [26]. Population disparities in the prevalence of GERD risk factors, such as rates of smoking, obesity, chronic diseases, dietary habits, or physical inactivity, explain the discrepancies in GERD prevalence between the Western Cape study and the current setting. The prevalence rate is comparable to rates reported using the GerdQ in other settings [1,37–39] but lower when compared with others [2,4,40,41].

The observed prevalence rates from the present setting translate to one in every three people, which is higher than the global rate of 1 in every six people [10,22,27,42]. Given the significant prevalence of Helicobacter pylori infection in the South African community: 54.5 percent [43] to 77.6 percent [44], the high rate of GERD from the present study appears contradictory. Research has shown that the elimination of Helicobacter pylori infection increased esophageal acid production and reflux, reduced esophageal peristalsis, and aggravated GERD symptoms in patients diagnosed with both Helicobacter pylori infection and GERD, suggesting the protective role of Helicobacter pylori against the occurrence of GERD [45]. The high rates of Helicobacter pylori infection in South Africa could suggest low rates of GERD, as in Nigeria,

**Table 2. Clinical factors associated with GERD among the adult population of Mthatha, South Africa.**

| Variables of interest | Categories | GERD, n (%) N = 113 | No GERD, n (%) N = 240 | Cramer's V | p-value |
|---|---|---|---|---|---|
| Nutritional Status | Obese | 44 (28.2) | 112 (71.8) | 0.128 | 0.124* |
| | Overweight | 29 (31.9) | 62 (68.1) | | |
| | Normal weight | 22 (31.0) | 49 (69.0) | | |
| | Underweight | 3 (50.0) | 3 (50.0) | | |
| | Morbid obesity | 15 (51.7) | 14 (48.3) | | |
| Diabetes mellitus | Yes | 12 (25.0) | 36 (75.0) | 0.060 | 0.340 |
| | No | 101 (33.1) | 204 (66.9) | | |
| Hypertension | Yes | 29 (43.9) | 37 (56.1) | 0.123 | 0.031 |
| | No | 84 (29.3) | 203 (70.7) | | |
| Hyperlipidemia | Yes | 4 (16.7) | 20 (83.3) | 0.089 | 0.149 |
| | No | 109 (33.1) | 220 (66.9) | | |
| Family History of GERD | Yes | 31 (43.7) | 40 (56.3) | 0.125 | 0.027 |
| | No | 82 (29.1) | 200 (70.9) | | |
| COPD | Yes | 12 (35.3) | 22 (64.7) | 0.023 | 0.812 |
| | No | 101 (31.7) | 218 (68.3) | | |
| Ischaemic Heart Disease | Yes | 0 (0.0) | 7 (100.0) | 0.098 | 0.154 |
| | No | 113 (32.7) | 233 (67.3) | | |

*Fisher's Exact.

a country with high rates of Helicobacter pylori infection but low GERD rates [24,46]. Despite a high rate of Helicobacter pylori infection, the high rate of GERD in the current population could be attributed to the less virulent phylogenies of the bacteria circulating in South Africa [44]. Although Helicobacter pylori infection status remains unknown in the current population, people suffering from peptic ulcers, who have been demonstrated to have a higher rate of this bacterium in the Mthatha population [43], were excluded.

Therefore, the observed high rates in the current setting could best be explained by demographic, health, and lifestyle differences within and across countries. Populations with more aging populations and population growth have observed a high prevalence of GERD [9]. In the present study, most persons with GERD were above 40. Aging increases the risk of GERD by altering the esophageal and esophagogastric junction mechano-physical properties, decreasing esophageal peristalsis, increasing esophageal acid exposure, or increasing the anatomical disruption of the oesophagogastric junction [47]. Moreover, the gender distribution could contribute to prevalence rates, mainly if there are more females, as with the current study. Female gender has been associated with a higher risk of GERD [25,48]. Also, high obesity rates, as in the present study, could support the high prevalence observed, a finding which corroborates with international [22,27] and national [25] literature.

Larger abdominal circumferences associated with obesity raise intra-abdominal pressure and the transdiaphragmatic gradient, which are linked to the development of GERD [49].

In addition, a high occurrence of hiatus hernia, which has been implicated in the pathogenesis of GERD, has been reported among overweight and obese persons [31,49].

Compared to GERD with minimal impact, the prevalence of GERD with high impact on everyday life was lower at 45 percent. This finding shows that GERD did not have much effect on the daily life of most of the participants in the present setting. Contrarily, more participants had GERD, significantly impacting daily life in Southwestern Saudi Arabia [4].

Heartburn was the most common symptom similar in Western countries [50], Jamaica [51], Brazil [52], Saudi Arabia [53], Taiwan [54], Indonesia [41], and Nigeria [24]. However, regurgitation was the most common symptom in other

**Table 3. Lifestyle factors associated with GERD among the adult population of Mthatha, South Africa.**

| Variables of interest | Categories | GERD, n (%) N = 113 | No GERD, n (%) N = 240 | Cramer's V | p-value |
|---|---|---|---|---|---|
| Smoking | Yes | 19 (29.2) | 46 (70.8) | 0.028 | 0.700 |
| | No | 94 (32.6) | 194 (67.4) | | |
| Alcohol consumption | Yes | 47 (43.5) | 61 (56.5) | 0.164 | 0.003 |
| | No | 66 (26.9) | 179 (73.1) | | |
| Sedentary lifestyle | Yes | 42 (29.2) | 102 (70.8) | 0.051 | 0.404 |
| | No | 71 (34.0) | 138 (66.0) | | |
| Health-conscious diet | Yes | 54 (27.7) | 141 (72.3) | 0.103 | 0.069 |
| | No | 59 (37.3) | 99 (62.7) | | |
| Diner to bedtime ≤2 hours | Yes | 57 (33.3) | 114 (66.7) | 0.027 | 0.688 |
| | No | 56 (30.8) | 126 (69.2) | | |
| Diner to bedtime >2 hours | Yes | 56 (30.8) | 126 (69.2) | 0.027 | 0.688 |
| | No | 57 (33.3) | 114 (66.7) | | |
| Walking less than 30 minutes per week | Never | 35 (46.7) | 40 (53.3) | 0.182 | 0.008 |
| | 1 time | 29 (23.6) | 94 (76.4) | | |
| | 2-3 times | 11 (28.2) | 28 (71.8) | | |
| | >3 times | 38 (33.0) | 77 (67.0) | | |
| Walking for 30 minutes or more per week | Never | 78 (28.2) | 199 (71.8) | 0.196 | 0.004 |
| | 1 time | 13 (65.0) | 7 (35.0) | | |
| | 2-3 times | 10 (41.7) | 14 (58.3) | | |
| | >3 times | 12 (38.7) | 19 (61.3) | | |
| Running less than 30 minutes per week | Never | 54 (28.1) | 138 (71.9) | 0.125 | 0.137 |
| | 1 time | 4 (25.0) | 12 (75.0) | | |
| | 2-3 times | 46 (36.5) | 80 (63.5) | | |
| | >3 times | 9 (50.0) | 9 (50.0) | | |
| Running 30 minutes per week | Never | 106 (32.0) | 225 (68.0) | 0.143 | 0.067 |
| | 1 time | 1 (20.0) | 4 (80.0) | | |
| | 2-3 times | 3 (23.1) | 10 (76.9) | | |
| | >3 times | 3 (100.0) | 0 (0.0) | | |
| Swimming for less than 30 minutes per week | Never | 103 (30.5) | 235 (69.5) | 0.179 | 0.003* |
| | 1 time | 2 (100.0) | 0 (0.0) | | |
| | >3 times | 8 (66.7) | 4 (33.3) | | |
| Swimming for 30 minutes per week | Never | 113 (32.1) | 239 (67.9) | N\C | |
| Exercise less than 30 minutes per week | Never | 102 (30.2) | 236 (69.8) | 0.245 | <0.001* |
| | 1 time | 11 (91.7) | 1 (8.3) | | |
| | 2-3 times | 0 (0.0) | 2 (100.0) | | |
| Exercise 30 minutes per week | Never | 102 (32.2) | 215 (67.8) | 0.114 | 0.216* |
| | 1 time | 9 (47.4) | 10 (52.6) | | |
| | 2-3 times | 1 (10.0) | 9 (90.0) | | |
| | >3 times | 1 (20.0) | 4 (80.0) | | |
| Alcohol consumption per week | Never | 66 (26.9) | 179 (73.1) | 0.277 | <0.001* |
| | 1 time | 34 (40.0) | 51 (60.0) | | |
| | 2-3 times | 3 (23.1) | 10 (76.9) | | |
| | >3 times | 10 (100.0) | 0 (0.0) | | |
| Number of meals a day | <3 meals | 28 (22.8) | 95 (77.2) | 0.305 | <0.001 |
| | 3 meals | 49 (28.3) | 124 (71.7) | | |
| | >3 meals | 36 (64.3) | 20 (35.7) | | |

Note: *Fisher's Exact. N\C = Not Computed.

**Table 4. Dietary factors associated with GERD among the adult population of Mthatha, South Africa.**

| Variables of interest | Categories | GERD, n (%) N = 113 | No GERD, n (%) N = 240 | Cramer's V | p-value |
|---|---|---|---|---|---|
| Citrus drinks | Yes | 53 (27.0) | 143 (73.0) | 0.122 | 0.030 |
| | No | 60 (38.5) | 96 (61.5) | | |
| Maize meal | Yes | 96 (31.9) | 205 (68.1) | 0.011 | 0.967 |
| | No | 17 (33.3) | 34 (66.7) | | |
| Milk | Yes | 82 (27.3) | 218 (72.7) | 0.245 | <0.001 |
| | No | 31 (59.6) | 21 (40.4) | | |
| Fast foods | Yes | 72 (31.6) | 156 (68.4) | 0.015 | 0.868 |
| | No | 41 (33.1) | 83 (66.9) | | |
| Spicy foods | Yes | 63 (32.6) | 130 (67.4) | 0.013 | 0.901 |
| | No | 50 (31.4) | 109 (68.6) | | |
| Greasy foods | Yes | 68 (36.2) | 120 (63.8) | 0.093 | 0.102 |
| | No | 45 (27.4) | 119 (72.6) | | |
| Fried foods | Yes | 79 (34.3) | 151 (65.7) | 0.066 | 0.263 |
| | No | 34 (27.9) | 88 (72.1) | | |
| High fat foods | Yes | 65 (32.0) | 138 (68.0) | 0.002 | 1.000 |
| | No | 48 (32.2) | 101 (67.8) | | |
| Grilled foods | Yes | 75 (31.3) | 165 (68.8) | 0.027 | 0.705 |
| | No | 38 (33.9) | 74 (66.1) | | |
| Meat | Yes | 98 (33.8) | 192 (66.2) | 0.078 | 0.187 |
| | No | 15 (24.2) | 47 (75.8) | | |
| Snacks | Yes | 82 (29.6) | 195 (70.4) | 0.103 | 0.073 |
| | No | 31 (41.3) | 44 (58.7) | | |
| Bread | Yes | 86 (28.6) | 215 (71.4) | 0.184 | 0.001 |
| | No | 27 (52.9) | 24 (47.1) | | |
| Vegetable | Yes | 91 (30.3) | 209 (69.7) | 0.091 | 0.122 |
| | No | 22 (42.3) | 30 (57.7) | | |
| Fruits | Yes | 103 (30.7) | 232 (69.3) | 0.129 | 0.031 |
| | No | 10 (58.8) | 7 (41.2) | | |
| Salt | Yes | 78 (30.4) | 179 (69.6) | 0.062 | 0.303 |
| | No | 35 (36.8) | 60 (63.2) | | |
| Sugar | Yes | 75 (28.8) | 185 (71.2) | 0.117 | 0.038 |
| | No | 38 (41.3) | 54 (58.7) | | |
| Chocolate | Yes | 82 (32.4) | 171 (67.6) | 0.011 | 0.943 |
| | No | 31 (31.3) | 68 (68.7) | | |
| Coffee | Yes | 64 (30.6) | 145 (69.4) | 0.041 | 0.517 |
| | No | 49 (34.5) | 93 (65.5) | | |
| Tea | Yes | 83 (30.9) | 186 (69.1) | 0.048 | 0.443 |
| | No | 30 (36.1) | 53 (63.9) | | |
| Soft drinks/Carbonated drinks | Yes | 62 (28.7) | 154 (71.3) | 0.092 | 0.109 |
| | No | 51 (37.5) | 85 (62.5) | | |
| Energetic drinks | Yes | 65 (33.9) | 127 (66.1) | 0.046 | 0.457 |
| | No | 47 (29.6) | 112 (70.4) | | |

**Table 5. Predictors of GERD among the adult population of Mthatha, South Africa.**

| Independent variables | Crude OR (95% CI) | Standard AOR (95%CI) | p-value | Firth AOR (95% CI) | p-value |
|---|---|---|---|---|---|
| Sex | | | | | |
| Female | 1.8 (1.0-3.0) | 1.6 (0.9-3.0) | 0.109 | 1.6 (0.9–2.9) | 0.116 |
| Male | Reference category | | | | |
| Hypertension | | | | | |
| Yes | 1.9 (1.1-3.3) | 3.2 (1.5-6.8) | 0.003 | 3.0 (1.5–6.4) | 0.003 |
| No | Reference category | | | | |
| Family History of GERD | | | | | |
| No | 0.5 (0.3-0.9) | 1.3 (0.6-2.8) | 0.572 | 1.2 (0.6–2.7) | 0.584 |
| Yes | Reference category | | | | |
| Alcohol consumption | | | | | |
| Yes | 2.1 (1.3-3.4) | 1.7 (0.9-3.0) | 0.080 | 1.6 (0.9–2.9) | 0.082 |
| No | Reference category | | | | |
| Number of meals a day | | | | | |
| 3 meals | 1.3 (0.8-2.3) | 0.8 (0.4-1.6) | 0.613 | 0.9 (0.5–1.6) | 0.630 |
| <3 meals | Reference category | | | | |
| Number of meals a day | | | | | |
| >3 meals | 6.1 (3.1-12.2) | 4.5 (2.0-10.0) | <0.001 | 4.2 (1.9–9.2) | <0.0001 |
| <3 meals | Reference category | | | | |
| Citrus drinks (No vs Yes) | | | | | |
| No | 1.7 (1.1-2.6) | 1.4 (0.8-2.5) | 0.216 | 1.4 (0.8–2.5) | 0.219 |
| Yes | Reference category | | | | |
| Milk | | | | | |
| No | 3.9 (2.1-7.2) | 1.9 (0.8-4.7) | 0.167 | 1.8 (0.8–4.3) | 0.180 |
| Yes | Reference category | | | | |
| Bread | | | | | |
| No | 2.8 (1.5-5.1) | 2.1 (0.9-5.1) | 0.085 | 2.1 (0.9–4.8) | 0.095 |
| Yes | Reference category | | | | |
| Fruits | | | | | |
| No | 3.2 (1.2-8.7) | 2.8 (0.9-8.5) | 0.073 | 2.6 (0.9–7.8) | 0.077 |
| Yes | Reference category | | | | |
| Sugar | | | | | |
| No | 1.7 (1.1-2.8) | 1.1 (0.5-2.1) | 0.845 | 1.1 (0.5–2) | 0.839 |
| Yes | Reference category | | | | |
| Walking for less than 30 minutes per week | | | | | |
| Yes | 2.2 (1.3-3.8) | 1.6 (0.8-3.3) | 0.223 | 1.7 (0.8–3.3) | 0.222 |
| No | Reference category | | | | |
| Swimming for less than 30 minutes per week | | | | | |
| Yes | 5.7 (1.7-18.6) | 3.6 (0.9-14.8) | 0.074 | 3.3 (0.9–13.4) | 0.075 |
| No | Reference category | | | | |
| Exercise less than 30 minutes per week | | | | | |
| Yes | 8.5 (2.3-31.1) | 3.9 (0.8-20.2) | 0.101 | 3.4 (0.8–17.4) | 0.112 |
| No | Reference category | | | | |

Hosmer and Lemeshow Test p-value = 0.266; Overall percentage = 77.8%; Nagelkerke $R^2$ = 30.1%; $X^2$ p-value <0.001; AOR = Adjusted Odd Ratio.

settings, including Southern Chile [55] and Saudi medical students [3]. Heartburn is the most common symptom among this study's participants and remains a classical GERD symptom [56]. However, GERD symptoms were mainly mild and related to the low impact on daily life observed in this study. Evidence suggests that not all reflux episodes are accompanied by symptoms, and heartburn frequently manifests as a sour aftertaste in the back of the mouth, either with or without refluxate regurgitation [56]. This could be why most people do not often report severe symptoms. The mild presentation of reflux symptoms in this population aligns with findings from the 24-hour ambulatory pH-metry study in the rural Eastern Cape community [32] and India [37]. Nonetheless, it is important to note that even subclinical reflux has a high propensity for serious long-term complications like Barrett's esophagus, a significant risk factor for esophageal cancer, which is one of the dominant cancers in this region. Indeed, radiological manifestations confirm the presence of esophageal stricture and esophageal mass in people with GERD and a suggestive diagnosis of esophageal adenocarcinoma and Barrett's esophagus in non-GERD persons presenting with intra-esophageal and extra-esophageal mild symptoms of GERD in the current setting [57]. These findings, together with the actual observation of the long-term complications of GERD, require proper monitoring and management of patients with mild GERD symptoms with little impact on daily life.

The present study found that having hypertension and having more than three meals per day were significant risk factors for GERD. The relationship between hypertension and GERD and silent GERD has been reported in previous research [58–60]. The observed relationship was more evident with untreated abdominal obesity and uncontrolled hypertension [59]. Although the use of calcium channel blockers showed no association [59], the association between hypertension and GERD has been explained by the effects of common anti-hypertensive drugs on the functioning of the lower esophageal sphincter. These drugs can enhance esophageal motor activity, increase esophageal acid exposure, reduce the tone of LES, or decrease esophageal clearance, thereby increasing GERD [61,62]. A recent systematic review suggested that GERD is a potential risk factor for hypertension [63], probably due to subjective perception of stress, poor lifestyle and dietary practices which has been identified as predictors of GERD [4]. The finding of the association between hypertension and GERD, and the shared predictors should direct public health interventions to reduce incidence as non-communicable diseases and cancer (including esophageal cancer a chronicity of GERD) are two prongs in the quadruple burden of disease in South Africa.

Concerning lifestyle factors, having more than three meals daily is associated with a higher risk of having GERD. The observed 4.2-fold risk with >3 meals/ day can be explained by disruption of the gastroesophageal barrier after eating large meals. Similar results were found where a significant relationship was reported between having 1–2 meals daily, having one big meal in the evening, and GERD [64]. Additionally, eating habits, including irregular meal patterns, eating many meals, and eating right before bed, may be linked to GERD symptoms [65]. Large meals can increase gastric distension and transient lower esophageal sphincter relaxations. This permits the stomach's contents to return to the esophagus [42,66].

The occurrence of GERD was not associated with other lifestyle factors such as alcohol consumption, indulging in shorter exercise or swimming sessions, although the findings revealed higher likelihood ratios. Contrarily, alcohol consumption has been linked to GERD [67], acting on multiple mechanisms including interfering with the anti-reflux barrier and impairing esophageal motility with resultant poor clearance of refluxate [68]. Again, indulging in physical activity for more than 30 minutes, three times a week, reduces the risk of developing GERD [69], contrary to the findings in the present study, probably due to the similar rate of physical inactivity among those with and without GERD.

Non-consumption of fruits, which has been identified elsewhere [70,71], had no significant association with the occurrence of GERD in the current population. The association between fruit consumption and GERD is an important aspect to discuss since the risk of having GERD is lower for those who consume the most fruits than those with the lowest intake [70]. Physiological mechanisms have been used to explain the underlying correlation between fruit consumption and GERD symptom reduction [70].

Physiologically, dietary nitrites are converted nonenzymatically into significant amounts of nitric oxide in the acidic gastric environment. Nitric oxide may encourage reflux because of its strong LES-relaxant effects [70].

Dietary fibers are known to scavenge stomach nitrites, reducing the substrate available to produce nonenzymatic nitric oxide [72]. Therefore, the absence or limited fruit consumption voids the stomach of dietary fibers to scavenge nitrites, thereby increasing the chances for the synthesis of nitric oxide, an effective LES relaxer [73,74]. Additionally, vitamins A and C, which can be sourced from fruits, prevent the development of GERD [75]. Another crucial dietary finding was the increased but not statistically significant likelihood of GERD with non-consumption of fiber-rich bread. Fiber-rich diet has been associated with a 14% decrease in risk of GERD [71] and reduced possibility of symptoms [71,76] The suggestive mechanism of high fiber intake and decreased risk of GERD include greater bowel movement frequency, which, decreases gastric stasis and the occurrence of reflux events [76], modulation of the gut microbiome, and influence on the gut-brain interactions [77]. These mechanisms increase the minimal lower esophageal sphincter resting pressure, decrease the number of gastroesophageal refluxes, and decrease GERD symptoms.

## Conclusions

The prevalence of GERD in Mthatha amounted to 32.0 percent, which is similar to those in other studies that used the same diagnostic tool. This study confirmed a higher prevalence of low-impact GERD compared to a lower prevalence of high-impact GERD among residents of Mthatha. The study found that heartburn was the most common GERD-related symptom. A propensity to have GERD was observed with low fiber and low fruit diets, as well as improper physical activities.

Cardiovascular disease, mainly hypertension, and lifestyle (more than three meals a day) contribute to the prevalence of GERD in this population. There is a need to screen for GERD among hypertensive patients and vice versa, since a significant risk association was observed between the two conditions. Since the literature noted that the risk is increased if hypertension is untreated and uncontrolled, there is a need for early identification and prompt treatment of hypertension in the general population. There is also a need for community education on the association between these two diseases and the probable pathways to protect against their occurrence, for example, by encouraging healthy dietary habits and proper physical activities.

## Limitations

This was the first research to investigate GERD's prevalence and identify significant risk factors in Mthatha, Eastern Cape province of South Africa. The study was, however, limited because of the facility-based design, reducing the validity of prevalence rates for the general population, due to the possibility of missing a subgroup of people with poor health-seeking behavior. Even though the sample size was adequate in estimating prevalence rates, the observed rates should be cited cautiously, considering the sampled population's demographic profile. In addition, recall bias could limit the study's findings because the participants were unaware of the survey before visiting the study site; they could have provided data that is not entirely a true reflection of real-life occurrences. Lastly, anti-hypertensive drug use was omitted, which could provide valuable insight since this factor remains contradictory in the current literature. Future researchers should consider measuring this variable to ascertain its relationship with GERD.

## Supporting information

**S1 File.**
(ZIP)

## Acknowledgments

We thank the management of Mthatha Regional Hospital for granting permission for this study to be carried out. We thank all the participants whose primary intention in visiting the health facility was not to take part in any research, yet they made a choice to do so. We also thank the Walter Sisulu University Directorate of Research and Innovation for the call to generate and disseminate findings from this work.

## Author contributions

**Conceptualization:** Nomonde Ndyalvan, Mirabel K. Nanjoh, Wezile W. Chitha, Sibusiso C. Nomatshila.

**Formal analysis:** Mirabel K. Nanjoh.

**Methodology:** Nomonde Ndyalvan, Mirabel K. Nanjoh, Monwabisi Faleni, Sibusiso C. Nomatshila.

**Project administration:** Nomonde Ndyalvan, Monwabisi Faleni.

**Supervision:** Mirabel K. Nanjoh, Sibusiso C. Nomatshila.

**Validation:** Mirabel K. Nanjoh, Wezile W. Chitha.

**Writing – original draft:** Mirabel K. Nanjoh.

**Writing – review & editing:** Nomonde Ndyalvan, Monwabisi Faleni, Wezile W. Chitha, Sibusiso C. Nomatshila.

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
