## [Decision Letter · Decision Letter 0]

20 Jun 2025

Dear Dr. Nanjoh,

Thank you for submitting your manuscript to PLOS ONE. After careful consideration, we feel that it has merit but does not fully meet PLOS ONE’s publication criteria as it currently stands. Therefore, we invite you to submit a revised version of the manuscript that addresses the points raised during the review process.

**ACADEMIC EDITOR: **

We look forward to receiving your revised manuscript.

Kind regards,

Emmanuel O Adewuyi, BPharm, MPH, PhD

Academic Editor

PLOS ONE

Journal Requirements:

Reviewers' comments:

Reviewer's Responses to Questions

**Comments to the Author**

1. Is the manuscript technically sound, and do the data support the conclusions?

Reviewer #1: Yes

Reviewer #2: No

Reviewer #3: Yes

Reviewer #4: Yes

Reviewer #5: Yes

Reviewer #6: Yes

2. Has the statistical analysis been performed appropriately and rigorously?

Reviewer #1: Yes

Reviewer #2: No

Reviewer #3: Yes

Reviewer #4: Yes

Reviewer #5: Yes

Reviewer #6: Yes

3. Have the authors made all data underlying the findings in their manuscript fully available?

Reviewer #1: Yes

Reviewer #2: No

Reviewer #3: Yes

Reviewer #4: No

Reviewer #5: Yes

Reviewer #6: No

4. Is the manuscript presented in an intelligible fashion and written in standard English?

Reviewer #1: No

Reviewer #2: Yes

Reviewer #3: Yes

Reviewer #4: Yes

Reviewer #5: Yes

Reviewer #6: Yes

Reviewer #1: Dear Authors, here are the comments:

1. Line 100 – Make it study design and period

2. Line 123 – “… selected using a sampling constant of six”, how was this interval calculated.

3. How did you use systematic sampling? What was the sampling frame?

4. Please revise the result and discussion part.

Reviewer #2: The manuscript presents a technically sound cross-sectional study on GORD prevalence and risk factors in the OR Tambo district. The use of the validated GerdQ tool for diagnosis and systematic sampling are strengths. However, several issues affect the robustness of conclusions:

Sampling Bias: Over 90% of participants were from one sub-district (King Sabata Dalindyebo), with Port St Johns unrepresented. This limits spatial generalizability and may skew risk factor associations.

Uncontrolled Confounders: The omission of anti-hypertensive medication use (noted in limitations) is critical, as these drugs (e.g., beta-blockers, calcium channel blockers, and ACE inhibitors) may independently influence GORD. This undermines the reported hypertension-GORD association (OR=3.5).

Risk Factor Analysis: The binary logistic regression model includes Coffee consumption despite non-significance (p=0.097) and an implausibly wide CI (OR=9.2; CI:0.7–127.9), suggesting model overfitting or sparse data. Re-evaluation with stricter variable selection is needed.

Symptom Interpretation: While heartburn was most prevalent (50.5%), mild symptoms dominated. The claim that GORD has "little effect on daily life" requires nuance, as subclinical reflux may still impact long-term complications (e.g., Barrett’s esophagus).

Conclusions are partially supported but overgeneralized. The high prevalence (32.3% vs. global 14–18%) is attributed to local risk factors, yet sampling bias and unmeasured confounders (e.g., H. pylori status, medication use) weaken causal inferences.

Statistical Rigor

Sample Size: Adequately calculated (n=406) for prevalence estimation but underpowered for subgroup analyses (e.g., dietary factors with small "no" groups, like fruit non-consumers: n=23).

Model Issues:

Hosmer-Lemeshow Test: Absent in results; essential for logistic regression validity.

Overfitting: Table 5 includes alcohol despite non-significance. A reduced model excluding non-significant variables is warranted.

Data Sparsity: Zero-cell counts (e.g., "swimming 30+ min/week": no GORD cases) may distort ORs. Sensitivity analyses (e.g., Firth correction) should be considered.

Inconsistencies:

Table 2 reports 45.9% GORD with family history (p=0.004), but this factor is excluded from the final regression model without justification.

Recommendations:

Address sampling bias by acknowledging reduced generalizability.

Re-run regression excluding non-significant variables and report model diagnostics.

Report effect sizes (e.g., Cramer’s V) for demographic trends.

Address data sparsity and consider exact logistic regression where applicable.

Recommendations for Revision:

Add Data Availability Statement.

Correct tables/typos and clarify demographic trend interpretations.

Discuss H. pylori’s protective role in context of South Africa’s high infection rates.

Reviewer #3: Summary:

This paper provides insights into the prevalence and risk factors of gastro-oesophageal reflux disease (GORD) in a rural South African population, an area with limited prior research. The findings are significant, particularly the high prevalence of GORD and its associations with hypertension, dietary habits, and lifestyle factors. However, the study has notable gaps and methodological limitations that need to be addressed before publication.

Comments:

Inclusion and Exclusion Criteria:

• The inclusion criteria specify adults aged 18 years and above, of both genders, seeking medical attention at the outpatient department of Mthatha Regional Hospital. However, the exclusion criteria mention individuals with selected comorbidities and pregnant women, but these comorbidities are not clearly defined.

• Suggestion: It would be helpful to clarify which comorbidities were excluded from the study to avoid ambiguity and ensure that readers understand the specific population being studied.

Address Missing Confounding Factors:

• The study should explore additional confounding variables that might influence the relationship between hypertension and GORD. For example, the impact of anti-hypertensive drugs, which could affect the lower oesophageal sphincter function, should be considered. It is also important to explore the influence of other medications or comorbid health conditions that could interact with GORD symptoms.

• Suggestion: A more thorough exploration of these confounding factors would strengthen the paper's scientific rigor.

Final Note:

This study is a valuable contribution to understanding GORD in rural South Africa. However, addressing the above points will enhance the paper’s scientific rigor, generalizability, and impact. Conditional acceptance with revisions is recommended to improve the clarity, methodological transparency, and applicability of the findings to a broader population.

Reviewer #4: Research Article Peer Review

Your research titled “Prevalence of gastro-oesophageal reflux disease and its associate risk factors among the adult population of OR Tambo district, South Africa” was reviewed and the following comments were made for your consideration.

Corresponding authorship discrepancies

• Mirabel Kah-Keh Nanjoh was indicated as the corresponding author in the form but the asterisk symbol was rather attached to Nomonde Ndyalvan in sentence 5. Could you please clarify?

• Sentence 18, please rectify the initials in the parenthesis, it should rather be “(N.N)”.

Title

• Please, “associate” should rather be “associated”.

• Please, “gastro-oesophageal” should rathe be “gastroesophageal”.

Abstract

Sentence 32, percentages of heartburn, sleep disturbance and regurgitation should not be reported in parenthesis to ensure consistency with similar results reported in the abstract.

Introduction

Even though the author indicated previous studies showcasing GORD prevalence globally and Africa, for instance, sentence 69, the author cited previous studies reporting GORD rates in west Africa. Nonetheless, the author could have included more similar work in African setting with country specific findings.

Materials and methods

• Sentence 123, sampling section. Could the author please clarify how he had the sampling interval of 6 for the systematic sampling?

• Sentence 133, “questionnaire which included validated questionnaires” could the author clarify this statement. Please how were the questionnaires validated? which statistical tool was used to check the questionnaire reliability?

• Sentence 139, “including comorbidities, WAS also collected” it should rather be “WERE”. Also, the author should please elaborate examples of comorbidities included in the study.

• Sentence 155, the author never site or state the standardized tools he used to collect data concerning physical activities, smoking, and exercise, could the author please elaborate?

Data Analysis

• Sentence 164, “Binary logistic regression's 95% confidence intervals and adjusted odd ratios were presented”. But I could not see any information concerning adjusted odds ratio in the results section of the binary logistic regression. Please elaborate.

• Sentence 165, “The Hosmer-Lemeshow test was used to assess the logistic regression model's goodness of fitness”. Please what was the score of the test? Please this can be reported as a footnote of the logistic regression table.

Results

• Please all tables’ headings are not complete and should be reviewed. For instance, “Table 1: Demographic factors associated with GORD”, Should rather be “Table 1: Demographic factors associated with GORD among adult population of OR Tambo district, South Africa”. Please note that, the target audience and study area should be included in the title of the tables. Please review and apply it throughout the protocol.

• Sentence 234, table 5. The author did not report on adjusted odds ratio. This means that, co-founders were not adjusted for in the model. Could the author please clarify?

• Sentence 234, table 5. Reference groups for the variables were not included in the table. For instance, hypertension (yes/no), which one is the reference group for the hypertension variable? Please review the table to include suggestions.

• Sentence 234, table 5, the table only included significant predictors, this makes the model highly problematic since co founders and non-significant predictors were not included and adjusted for in the model. Please review to include demographics, age sex among others.

References

• Please spaces between individual references are wider. Please reformat the reference list to reduce the spaces.

General comments

• To prevent unbiased sampling procedure, the author should have given equal chance for all health facilities in the district to have been selected for the study, instead, the author purposely selected a single health facility. This method is biased and may affect the study findings to be regarded as the true reflection of the study population. Multi stage probability sampling should have been employed in this study to reduce sampling bias.

• Response to data availability unclear. Could the author please clarify?.

Reviewer #5: The Manuscript fits into the scope and its recommended for publication.

It is technically sound based on the methodology with data which reflects on the study . Data presented are appropriately based on fact.

Reviewer #6: I find your manuscript to be well-organized, clearly written, and methodologically sound. You have provided a coherent and logical rationale for all major components of the study, including the research objectives, study design, data collection methods, and analytical approaches. This thoughtful justification strengthens the internal consistency of the manuscript and enhances its scientific credibility.

The study addresses a relevant topic, and the findings are well presented. The discussion appropriately contextualizes the results within the existing body of literature, and the limitations are reasonably acknowledged.

**Do you want your identity to be public for this peer review?** For information about this choice, including consent withdrawal, please see our Privacy Policy

Reviewer #1: No

Reviewer #2: No

Reviewer #3: No

Reviewer #4: **Yes: ** Hardi Adam

Reviewer #5: **Yes: ** Ebenezer Ad Adams

Reviewer #6: No

---

## [Author Response · Author response to Decision Letter 1]

12 Aug 2025

Authors’ Note:

Dear reviewers,

Greetings and thank you for all the valuable grammatical, scientific, and statistical inputs. Based on the concerns raised on the sub-district distribution of the sampled population, the revised findings presented only for Mthatha; the setting that made up more than made up 89% of the population of the original manuscript. The title was changed to reflect the findings. All concerns were taken into consideration in re-analyzing and re-writing the manuscript. The original manuscript with tracked changes and the clean updated manuscript are attached.

Please find the responses to the comments. Thank you

Reviewer #1: Dear Authors, here are the comments:

1. Line 100 – Make it study design and period

Response: It has been made Study design and period as in line 97

2. Line 123 – “… selected using a sampling constant of six”, how was this interval calculated.

Response: Calculation explained as in line 127-128

3. How did you use systematic sampling? What was the sampling frame?\

Response: Sample frame statement as in line 123-125

4. Please revise the result and discussion part.

Response: Results and discussion sections has been revised

Reviewer #2: The manuscript presents a technically sound cross-sectional study on GORD prevalence and risk factors in the OR Tambo district. The use of the validated GerdQ tool for diagnosis and systematic sampling are strengths. However, several issues affect the robustness of conclusions:

Sampling Bias: Over 90% of participants were from one sub-district (King Sabata Dalindyebo), with Port St Johns unrepresented. This limits spatial generalizability and may skew risk factor associations.

Response: Yes, very correct. The revised manuscript is center on data for Mthatha only.

Uncontrolled Confounders: The omission of anti-hypertensive medication use (noted in limitations) is critical, as these drugs (e.g., beta-blockers, calcium channel blockers, and ACE inhibitors) may independently influence GORD. This undermines the reported hypertension-GORD association (OR=3.5).

Response: Agreed and was acknowledged as a limitation.

Risk Factor Analysis: The binary logistic regression model includes Coffee consumption despite non-significance (p=0.097) and an implausibly wide CI (OR=9.2; CI:0.7–127.9), suggesting model overfitting or sparse data. Re-evaluation with stricter variable selection is needed.

Response: Thank you for the comment, I had a look and Coffee consumption was not included in the regression model. The variable with the stated statistical values was Alcohol consumption. The wide CI was a typo (bulleted numbering). The adjusted OR for alcohol consumption was however, not statistically significant (p=0.097).

There might be slight changes in these statistics due to a change in the sampled population.

Symptom Interpretation: While heartburn was most prevalent (50.5%), mild symptoms dominated. The claim that GORD has "little effect on daily life" requires nuance, as subclinical reflux may still impact long-term complications (e.g., Barrett’s esophagus).

Response: Thank you for the comment. This was taken into account and added to the discussion as in Line 311-320.

Conclusions are partially supported but overgeneralized. The high prevalence (32.3% vs. global 14–18%) is attributed to local risk factors, yet sampling bias and unmeasured confounders (e.g., H. pylori status, medication use) weaken causal inferences.

Response:

Sampling bias is minimal as only a single locality was re-analyzed and reported. Subgroup small size was corrected using the Firth Logistic regression analysis.

Participants with high H. pylori rate in the Mthatha locality (peptic ulcer) were excluded.

Medication use (anti-hypertensive drugs) was acknowledged as a limitation.

Statistical Rigor

Sample Size: Adequately calculated (n=406) for prevalence estimation but underpowered for subgroup analyses (e.g., dietary factors with small "no" groups, like fruit non-consumers: n=23).

Response: Firth logistics regression odds ratios were reported.

Model Issues:

Hosmer-Lemeshow Test: Absent in results; essential for logistic regression validity.

Response: Hosmer-Lemeshow Test significance stated as in Table 5 line 254 and in line 183

Overfitting: Table 5 includes alcohol despite non-significance. A reduced model excluding non-significant variables is warranted.

Response: Alcohol consumption was actually significant in the previous (Table 3: p=0.035) and even the current submission (Table 3: p=0.003) line 237.

Data Sparsity: Zero-cell counts (e.g., "swimming 30+ min/week": no GORD cases) may distort ORs. Sensitivity analyses (e.g., Firth correction) should be considered.

Response: Thank you for not only pointing out this error but for informing us how it can be rectified. This was very helpful and was considered in the current analysis. The Firth logistic regression outputs are presented in Table 5 line 254

Inconsistencies:

Table 2 reports 45.9% GORD with family history (p=0.004), but this factor is excluded from the final regression model without justification.

Response: Family history was added, but only significant variables and those showing neutral association were displayed in the table.

Recommendations:

Address sampling bias by acknowledging reduced generalizability.

Response: Done line 395-398

Re-run regression excluding non-significant variables and report model diagnostics.

Response: Done. All variables entered into the model appear in Table 5. These were significant in the bivariate analysis

Report effect sizes (e.g., Cramer’s V) for demographic trends.

Response: Done. Cramer’s V values were computed and reported in Tables 1-4.

Address data sparsity and consider exact logistic regression where applicable.

Response: Done, as previously explained. Thank you

Recommendations for Revision:

Add Data Availability Statement.

Response: Data availability statement added to the revised manuscript as in line 412 - 416

Correct tables/typos and clarify demographic trend interpretations.

Response: Tables and Typo corrected.

Discuss H. pylori’s protective role in context of South Africa’s high infection rates.

Response: The role of H. pylori discussed as in line 264-278.

Reviewer #3: Summary:

This paper provides insights into the prevalence and risk factors of gastro-oesophageal reflux disease (GORD) in a rural South African population, an area with limited prior research. The findings are significant, particularly the high prevalence of GORD and its associations with hypertension, dietary habits, and lifestyle factors. However, the study has notable gaps and methodological limitations that need to be addressed before publication.

Comments:

Inclusion and Exclusion Criteria:

• The inclusion criteria specify adults aged 18 years and above, of both genders, seeking medical attention at the outpatient department of Mthatha Regional Hospital. However, the exclusion criteria mention individuals with selected comorbidities and pregnant women, but these comorbidities are not clearly defined.

Response: Thank you for the comments. Comorbidities defined as in line 119-121.

• Suggestion: It would be helpful to clarify which comorbidities were excluded from the study to avoid ambiguity and ensure that readers understand the specific population being studied.

Response: Comorbidities specified as in line 119-121.

Address Missing Confounding Factors:

• The study should explore additional confounding variables that might influence the relationship between hypertension and GORD. For example, the impact of anti-hypertensive drugs, which could affect the lower oesophageal sphincter function, should be considered. It is also important to explore the influence of other medications or comorbid health conditions that could interact with GORD symptoms.

• Suggestion: A more thorough exploration of these confounding factors would strengthen the paper's scientific rigor.

Response: Done. Data on the use of anti-hypertensive drugs was not obtained, and this has been acknowledged as a limitation. The study is a first in this setting and as such, the parameters measured in this study was based on literature from other settings at the time of literature review. The strengths is that the influence of these drug on GERD is inconclusive with some studies affirming while others refute any associations. Nonetheless, the findings of this study and the design will greatly guide future research

Final Note:

This study is a valuable contribution to understanding GORD in rural South Africa. However, addressing the above points will enhance the paper’s scientific rigor, generalizability, and impact. Conditional acceptance with revisions is recommended to improve the clarity, methodological transparency, and applicability of the findings to a broader population.

Response: Thank you.

Reviewer #4: Research Article Peer Review

Your research titled “Prevalence of gastro-oesophageal reflux disease and its associate risk factors among the adult population of OR Tambo district, South Africa” was reviewed and the following comments were made for your consideration.

Corresponding authorship discrepancies

• Mirabel Kah-Keh Nanjoh was indicated as the corresponding author in the form but the asterisk symbol was rather attached to Nomonde Ndyalvan in sentence 5. Could you please clarify?

Response: Thank you for picking up this discrepancy. It has been rectified. Mirabel K. Nanjoh is the correct corresponding author. Line 4

• Sentence 18, please rectify the initials in the parenthesis, it should rather be “(N.N)”.

Title

Response: Thank you for picking up this discrepancy. It has been rectified. The corresponding author, however, has been replaced in accordance with the previous response. Line 16

• Please, “associate” should rather be “associated”.

Response: Thank you. Spelling changed as proposed line 1

• Please, “gastro-oesophageal” should rathe be “gastroesophageal”.

Response: Spelling changed as proposed throughout the manuscript.

Abstract

Sentence 32, percentages of heartburn, sleep disturbance and regurgitation should not be reported in parenthesis to ensure consistency with similar results reported in the abstract.

Response: Done Line 26-30

Introduction

Even though the author indicated previous studies showcasing GORD prevalence globally and Africa, for instance, sentence 69, the author cited previous studies reporting GORD rates in west Africa. Nonetheless, the author could have included more similar work in African setting with country specific findings.

Response: Thank you, country specific rates are stated as line 65-67

Materials and methods

• Sentence 123, sampling section. Could the author please clarify how he had the sampling interval of 6 for the systematic sampling?

Response: Sampling interval calculation as in line 126-128

• Sentence 133, “questionnaire which included validated questionnaires” could the author clarify this statement. Please how were the questionnaires validated? which statistical tool was used to check the questionnaire reliability?

Response: Thank you for pointing this out. The said validated questionnaire referred to GerdQ. However, the current questionnaire was acceptable at a Cronbach alpha value of 77.6%.

• Sentence 139, “including comorbidities, WAS also collected” it should rather be “WERE”. Also, the author should please elaborate examples of comorbidities included in the study.

Response: Comorbidities added in the questionnaire are stated in line 149-150

• Sentence 155, the author never site or state the standardized tools he used to collect data concerning physical activities, smoking, and exercise, could the author please elaborate?

Response: The variables measured were obtained from published literature and these are cited line 167-168

Data Analysis

• Sentence 164, “Binary logistic regression's 95% confidence intervals and adjusted odd ratios were presented”. But I could not see any information concerning adjusted odds ratio in the results section of the binary logistic regression. Please elaborate.

Response: Done- The binary logistic regression reported adjusted odd ratio. Adjusted was omitted in the table.

• Sentence 165, “The Hosmer-Lemeshow test was used to assess the logistic regression model's goodness of fitness”. Please what was the score of the test? Please this can be reported as a footnote of the logistic regression table.

Response: Thank you, Score, added line 183 and line 254 (Table 5)

Results

• Please all tables’ headings are not complete and should be reviewed. For instance, “Table 1: Demographic factors associated with GORD”, Should rather be “Table 1: Demographic factors associated with GORD among adult population of OR Tambo district, South Africa”. Please note that, the target audience and study area should be included in the title of the tables. Please review and apply it throughout the protocol.

Response: Thanks a lot for the clarity. The target audience and study area added to all Table labels.

• Sentence 234, table 5. The author did not report on adjusted odds ratio. This means that, co-founders were not adjusted for in the model. Could the author please clarify?

Response: Co-founders were adjusted. The model included all variables with significant crude odd ratios in the bivariate analysis.

• Sentence 234, table 5. Reference groups for the variables were not included in the table. For instance, hypertension (yes/no), which one is the reference group for the hypertension variable? Please review the table to include suggestions.

Response: Thank you. Done. Reference group indicated in Table 5

• Sentence 234, table 5, the table only included significant predictors, this makes the model highly problematic since co founders and non-significant predictors were not included and adjusted for in the model. Please review to include demographics, age sex among others.

Response: Thank you. All significant bivariate variables are shown in Table 5.

References

• Please spaces between individual references are wider. Please reformat the reference list to reduce the spaces.

Response: Thank you, spaces between references removed.

General comments

• To prevent unbiased sampling procedure, the author should have given equal chance for all health facilities in the district to have been selected for the study, instead, the author purposely selected a single health facility. This method is biased and may affect the study findings to be regarded as the true reflection of the study population. Multi stage probability sampling should have been employed in this study to reduce sampling bias.

Response: Sampled population readjusted. The data was re-analysed with only data from one locality.

• Response to data availability unclear. Could the author please clarify?

Response: Data availability statement added.

Reviewer #5: The Manuscript fits into the scope and its recommended for publication.

It is technically sound based on the methodology with data which reflects on the study . Data presented are appropriately based on fact.

Response: Thank you.

Reviewer #6: I find your manuscript to be well-organized, clearly written, and methodologically sound. You have provided a coherent and logical rationale for all major components of the study, including the research objectives, study design, data collection methods, and analytical approaches. This thoughtful justification strengthens the internal consistency of the manuscript and enhances its scientific credibility.

Response: Thank you.

The study addresses a relevant topic, and the findings are well presented. The discussion appropriately contextualizes the results within the existing body of literature, and the limitations are reasonably acknowledged.

Response: Thank you.

---

## [Decision Letter · Decision Letter 1]

29 Sep 2025

Dear Dr. Nanjoh,

Thank you for submitting your manuscript to PLOS ONE. After careful consideration, we feel that it has merit but does not fully meet PLOS ONE’s publication criteria as it currently stands. Therefore, we invite you to submit a revised version of the manuscript that addresses the points raised during the review process.

**ACADEMIC EDITOR: ** Minor revision has been recommended by reviewers. Author need to pay close attention to the required corrections, including those from the reviewer recommending acceptance. This revision also provides authors the opportunity to thoroughly proofread their work, and fact-check every statement so there is no delay moving from this point.

We look forward to receiving your revised manuscript.

Kind regards,

Emmanuel O Adewuyi, BPharm, MPH, PhD

Academic Editor

PLOS ONE

Journal Requirements:

Reviewers' comments:

Reviewer's Responses to Questions

**Comments to the Author**

Reviewer #1: (No Response)

Reviewer #3: All comments have been addressed

Reviewer #4: All comments have been addressed

Reviewer #6: All comments have been addressed

2. Is the manuscript technically sound, and do the data support the conclusions?

Reviewer #1: Yes

Reviewer #3: Yes

Reviewer #4: Yes

Reviewer #6: Yes

3. Has the statistical analysis been performed appropriately and rigorously?

Reviewer #1: Yes

Reviewer #3: Yes

Reviewer #4: Yes

Reviewer #6: Yes

4. Have the authors made all data underlying the findings in their manuscript fully available?

Reviewer #1: Yes

Reviewer #3: Yes

Reviewer #4: Yes

Reviewer #6: No

5. Is the manuscript presented in an intelligible fashion and written in standard English?

Reviewer #1: No

Reviewer #3: Yes

Reviewer #4: Yes

Reviewer #6: Yes

Reviewer #1: Dear Authors, All the comments have not been addressed. Here are the comments that need to be addressed:

1. Line 123 – “… selected using a sampling constant of six”, how was this interval calculated.

2. How did you use systematic sampling? What was the sampling frame?

3. Please revise the result and discussion part.

Reviewer #3: My earlier concerns regarding unclear inclusion and exclusion criteria, the absence of clarity around comorbidities, and the need to better acknowledge potential confounding factors have been adequately addressed in the revised manuscript. The authors have clarified the study population, defined comorbidities, acknowledged the limitations around unmeasured variables such as anti-hypertensive drug use, and re-analyzed the data to focus appropriately on Mthatha. These revisions have improved the methodological transparency and strengthened the manuscript.

Reviewer #4: I am satisfied with the revisions made by the authors. The data underlying the findings have been made fully available, and the manuscript is clearly presented in standard English. The work is technically sound, with the data appropriately supporting the conclusions. Furthermore, the statistical analyses have been conducted rigorously and appropriately. I therefore find the manuscript suitable for acceptance.

Reviewer #6: The manuscript in its current form is suitable for publication as all comments have been addressed. However, please replace the symbol ‘@’ in the Table 3 with a clearer notation such as ‘N\C’ (Not Computed). For clarity, you may also include a footnote below the table, for example:

Note: N\C = Not Computed.

**Do you want your identity to be public for this peer review?** For information about this choice, including consent withdrawal, please see our Privacy Policy

Reviewer #1: No

Reviewer #3: No

Reviewer #4: **Yes: ** Hardi Adam

Reviewer #6: No

---

## [Author Response · Author response to Decision Letter 2]

21 Oct 2025

Authors’ Note:

Dear reviewers,

Greetings and thank you for all the additional comments to strengthen the quality of the manuscript.

Please find the responses to the comments. Thank you

Reviewer #1: Dear Authors, All the comments have not been addressed. Here are the comments that need to be addressed:

1. Line 123 – “… selected using a sampling constant of six”, how was this interval calculated.

Response: Thank you for this comment. The interval was calculated by dividing the population size (N= 2500) by the estimated sample size (n=406) ~6.2. Please verify if Line 126 to Line 128 of the manuscript addresses this comment.

2. Reviewer #1: How did you use systematic sampling?

Response: Thank you for requesting clarity on this important component on systematic sampling. In this study, it began with a random selection of the first participant on each data collection day, then every sixth person. Please verify if Line 125 to Line 126 of the manuscript addresses this comment.

Reviewer #1: What was the sampling frame?

Response: Thank you for requesting clarity on this important component on systematic sampling. The sampling frame was the total number of people who will be available (expected to visit the outpatient department of the hospital) during the one-month data collection period (N= 2500). Please verify if Line 123 to Line 125 of the manuscript addresses this comment.

Reviewer #1: 3. Please revise the result and discussion part.

Response: Thank you for highlighting the need to revise the results and discussion part. This has been done. Please refer to Table 3 and Line 239.

Reviewer #3: My earlier concerns regarding unclear inclusion and exclusion criteria, the absence of clarity around comorbidities, and the need to better acknowledge potential confounding factors have been adequately addressed in the revised manuscript. The authors have clarified the study population, defined comorbidities, acknowledged the limitations around unmeasured variables such as anti-hypertensive drug use, and re-analyzed the data to focus appropriately on Mthatha. These revisions have improved the methodological transparency and strengthened the manuscript.

Response: Thank you. We highly appreciate your comment.

Reviewer #4: I am satisfied with the revisions made by the authors. The data underlying the findings have been made fully available, and the manuscript is clearly presented in standard English. The work is technically sound, with the data appropriately supporting the conclusions. Furthermore, the statistical analyses have been conducted rigorously and appropriately. I therefore find the manuscript suitable for acceptance.

Response: Thank you. We highly appreciate your comment.

Reviewer #6: The manuscript in its current form is suitable for publication as all comments have been addressed. However, please replace the symbol ‘@’ in the Table 3 with a clearer notation such as ‘N\C’ (Not Computed). For clarity, you may also include a footnote below the table, for example:

Note: N\C = Not Computed.

Response: Thank you for this quality improvement recommendation for the results section. The recommended input was considered. Please refer to Table 3 and Line 239.

---

## [Editor Report · Decision Letter 2]

4 Nov 2025

Prevalence of gastroesophageal reflux disease and its associated risk factors among the adult population of Mthatha, Eastern Cape, South Africa

PONE-D-25-16621R2

Dear Dr. Nanjoh,

We’re pleased to inform you that your manuscript has been judged scientifically suitable for publication and will be formally accepted for publication once it meets all outstanding technical requirements.

Kind regards,

Emmanuel O Adewuyi, BPharm, MPH, PhD

Academic Editor

PLOS ONE
---

## [Editor Report · Acceptance letter]

PONE-D-25-16621R2

PLOS ONE

Dear Dr. Nanjoh,

I'm pleased to inform you that your manuscript has been deemed suitable for publication in PLOS ONE. Congratulations! Your manuscript is now being handed over to our production team.

Kind regards,

on behalf of

Dr. Emmanuel O Adewuyi

Academic Editor

PLOS ONE